# Hierarchical Reinforcement Learning for Crude Oil Supply Chain Scheduling

**Nan Ma [1], Ziyi Wang [2,3], Zeyu Ba [2,3], Xinran Li [2,3], Ning Yang [2,3],\*, Xinyi Yang [1]**  **and Haifeng Zhang [2,3]**

[1] Key Laboratory of Oil & Gas Business Chain Optimization, CNPC, Petrochina Planning and Engineering Institute, Beijing 100083, China; manan2013@petrochina.com.cn (N.M.); yangxy234@petrochina.com.cn (X.Y.)

[2] Institute of Automation, Chinese Academy of Sciences, Beijing 100190, China; wangziyi2021@ia.ac.cn (Z.W.); bazeyu2022@ia.ac.cn (Z.B.); lixinran2022@ia.ac.cn (X.L.); haifeng.zhang@ia.ac.cn (H.Z.)

[3] School of Artificial Intelligence, University of Chinese Academy of Sciences, Beijing 100190, China

\* Correspondence: ning.yang@ia.ac.cn; Tel.: +86-1305-150-9251

**Abstract:** Crude oil resource scheduling is one of the critical issues upstream in the crude oil industry chain. It aims to reduce transportation and inventory costs and avoid alerts of inventory limit violations by formulating reasonable crude oil transportation and inventory strategies. Two main difficulties coexist in this problem: the large problem scale and uncertain supply and demand. Traditional operations research (OR) methods, which rely on forecasting supply and demand, face significant challenges when applied to the complicated and uncertain short-term operational process of the crude oil supply chain. To address these challenges, this paper presents a novel hierarchical optimization framework and proposes a well-designed hierarchical reinforcement learning (HRL) algorithm. Specifically, reinforcement learning (RL), as an upper-level agent, is used to select the operational operators combined by various sub-goals and solving orders, while the lower-level agent finds a viable solution and provides penalty feedback to the upper-level agent based on the chosen operator. Additionally, we deploy a simulator based on real-world data and execute comprehensive experiments. Regarding the alert number, maximum alert penalty, and overall transportation cost, our HRL method outperforms existing OR and two RL algorithms in the majority of time steps.

**Keywords:** hierarchical reinforcement learning; crude oil supply chain scheduling; operations research; linear programming

## 1. Introduction

With the rapid development of the economy and society, the production scheduling problem has gained more attention than ever in various industries. In particular, resolving the scheduling optimization with limited resources in real-world scenarios has become crucial [1–3]. In the field of petroleum energy, the resource production scheduling of crude oil is a sequential process of the industry chain. From upstream to downstream, there are seven processes in this complex industrial chain, including petroleum production, procurement, refining, sales, product transportation, storage, and trade. These aspects are closely linked and influence each other [4]. However, in recent years, unexpected events and uncertainties in international oil prices, geopolitics, and foreign and domestic markets have continuously made the steady operation of the petroleum industry chain more challenging. Moreover, articulation conflicts caused by local supply and demand imbalances have been common [5]. Therefore, in such an uncertain environment, an efficient and accurate petroleum resource scheduling decision is an effective way to solve this problem [6].

The crude oil supply chain scheduling optimization problem can be summarized as a resource optimization allocation problem of a multi-level supply chain complex system. Previous research on this issue has focused on OR, complex system science, and manage-

ment and has accumulated specific results [7–9]. However, due to the inherent complexity of this issue, there are still many challenges:

First, current optimization research and applications of petroleum resource scheduling are mainly at the long-period strategic level, which usually takes months or years. Since this period is relatively long, supply and demand will change continuously, affected by the actual environment. However, the traditional OR method relies on the accurate prediction of supply and demand to realize the modeling and solution of scheduling problems [10]. Consequently, it is difficult to precisely predict supply and demand in long-term decision-making, which limits traditional methods of solving such problems.

Second, due to uncertain factors in the entire petroleum industry chain, the results obtained by OR are not good enough. Uncertainties such as node inventory, transportation on the road, node demand, and future demand will all lead to a dynamically changing environment. However, in the scheduling field, traditional OR methods are only suitable for solving model-based problems with a deterministic state transfer distribution. Regarding highly uncertain environments, OR has a gap in performance in terms of adaptability and efficiency [11].

Third, the OR method will encounter many difficulties when solving the petroleum scheduling problem, which is a large-scale problem [12,13]. Due to the increasing dimension of large-scale problems, it is time-consuming to explore the vast decision space effectively. On the other hand, high dimensionality will also increase the number of locally optimal solutions to the problem, making it difficult for the algorithm to find the global optimal solution. In addition, concerning non-convex and non-differentiable cases, classical algorithms cannot be applied to large-scale problems.

To solve these three main challenges, our work optimizes a problem with random supply and demand. This problem differs from the preview work to optimize a scenario where all information is priorly known. (1) It is challenging to accurately predict supply and demand in long-period decision-making; thus, traditional methods are not applicable. We establish a model to solve time-varying environments which change every day. (2) To meet the challenges of uncertain factors in a dynamic environment, we use RL to learn an optimization policy through the online interaction between the agent and the environment. (3) Regarding crude oil supply chain scheduling being a large-scale problem, our work uses HRL to solve the high-dimensional problem. HRL decomposes the complex problem into several sub-problems and divides the strategy into different levels of sub-strategies [14]. Although it performs well in solving large-scale problems, HRL still needs more reasoning ability and better representation efficiency.

The intelligent scheduling of the crude oil supply chain needs to predictably generate dynamic adjustment plans when facing various uncertain factors and emergencies and use them as the basis for contingency plans. In this paper, we combine HRL with traditional optimization algorithms to provide a fast solution to the intelligent scheduling problem of the crude oil supply chain, and the main contributions include:

- A Novel Hierarchical Optimization Framework: The framework uses an upper level to control the dynamic impacts of uncertain environmental variables; thus, the large-scale long-period scheduling optimization problem is broken down into small-scale single-step problems. To the best of our knowledge, this framework is first proposed and used in the supply chain scheduling problem.
- A Well-designed HRL Algorithm: We formulate the Markov Decision Process (MDP) for the supply chain scheduling problem. We use RL as the upper-level agents that determine the desired inventory volume at the end of the time step. The lower-level agent conducts a quick local search and satisfies the inventory limits while sending penalty feedback to the upper-level agent. The strengths of both methods are subtly integrated.
- Deployment: We deploy our algorithm to a large-scale long-period crude oil scheduling optimization problem of CNPC. We develop a simulator with real-world data and

demonstrate that our approach can significantly improve stable inventory volumes and low transportation costs.

The remainder of the paper is organized as follows. In Section 2, we briefly introduce the fundamental problems and HRL, which are the main frameworks in our method. Section 3 formally formulates the crude oil supply chain scheduling issue. In Section 4, we illustrate the overall architecture of the HRL scheduling scheme. Section 5 introduces the dataset and the simulator we used to evaluate and train RL algorithms. In Section 6, we implement the experiment and discuss comparing baselines and our method. Finally, in Section 7, we draw the conclusions.

## 2. Related Work

This section introduces the fundamental problems and summarizes the crude oil supply chain scheduling problem as a resource allocation problem. In addition, we also present the main hierarchical idea used in our method.

### 2.1. Crude Oil Supply Chain Scheduling Issues

In the past few decades, the most popular and effective models for describing the form and function of supply chains have been based on mathematical programming. They were also combined with heuristic algorithms for optimization [15–19]. Mohammad [16] proposed a mixed integer linear programming and two approaches to minimize the total weighted tardiness and transportation costs. These two approaches include an exact procedure based on a Branch-and-Bound (B&B) algorithm and a metaheuristic genetic algorithm (GA). The B&B algorithm performs well in solving small-scale problems, but its running speed is slower than GA, while GA performs well in solving large-scale problems. Nima [17] established a mathematical model for multi-objective scheduling problems and customized two meta-heuristics algorithms, including the multi-objective particle swarm optimization algorithm and the non-dominated sorting genetic algorithm to minimize the total weighted tardiness and the total operation time. This work gained good results on both small and large-scale problems. However, it lacks constraints for real-life conditions, leading to a low transfer ability. Considering the uncertainties in intelligent production, Thitipong [18] used a strategy that integrates event-driven and period-driven methods to minimize the makespan. This method considered a dynamic environment but only fit in some industry scenarios, lacking versatility. Ali [19] established a mathematical model of bi-objective linear programming to minimize logistical costs and incorporated possible scenarios and fuzzy data, increasing the flexibility of the system. This method only considered uncertain factors in a simplified environment, which might not be suitable for complex environments involving more effect factors and constraints.

### 2.2. Complex Resource Allocation Approaches

Resource scheduling is a common problem in OR and production. An efficient scheduling plan can save resources and gain benefits. We model the crude oil supply chain scheduling problem as a resource scheduling problem for analysis. Considering some system dynamics model parameters are indeterminate, Mu [20] proposed a data-driven optimization approach to cope with multi-period resource allocation problems with periodic incoming observations. This approach has scenario-based stochastic planning which cannot solve large-scale problems, e.g., real-world decisions. Mathematical programming models are standard in solving supply chain problems, including mixed integer linear and nonlinear programming [21]. In multi-objective optimization problems, modeling methods simultaneously optimizing individual and overall objectives are effective [22]. Prior works have already applied methods based on RL to resolve dynamic resource allocation problems in real-life situations, including mobile edge computing [23,24], vehicle network traffic distribution [25], and predictive production planning for large manufacturing systems [26]. The resource scheduling problem has the properties of multi-constraints, computational

complexity, uncertainty, and multi-objectives. Since problem modeling is usually relatively complicated, it is necessary to introduce new methods to solve it.

*2.3. HRL*

Traditional RL methods often suffer from dimensional disasters when dealing with large-scale problems. Specifically, when the environment or task is relatively complex, the state space of the agent will be vast. This considerable state space will rapidly increase the parameters learned and the required storage space, making it difficult for RL to achieve the desired effect. To solve this problem, researchers propose HRL, of which the basic idea is to divide the problem into multiple levels. The upper layer calls the lower layer to solve the task, and the lower layer executes the command of the upper layer.

Currently, HRL is mainly divided into goal-based and option-based, where the goal-based method is more mainstream than the other. For instance, Nachum [27] chose the state space as the objective to solve the off-policy of HRL in the non-stationary problem. To approximate the solution of combinatorial optimization problems with time windows, a hierarchical strategy is employed to find the optimal solution under constraints [28]. Each layer of the hierarchy is designed with a separate reward function, enabling regular training. Hierarchical decision-making approaches can also significantly reduce the reliance on large amounts of labeled data [29] by learning sub-policies for each action and a master policy for the policy selection. There is less related work based on options, which can be observed as a summary of actions. Bacon [30] developed the option–critic algorithm by combining the options framework and deep RL. Here, the option is a sub-policy, and each option corresponds to a lower-layer policy. In the scheduling and resource allocation problem, Ren [31] proposed the hierarchical trajectory optimization and offloading optimization, which decomposed the scheduling problem into two layers of sub-problems and performed alternate optimization by HRL. Ying He [32] combined HRL and meta-learning, which made it possible to quickly adapt to new tasks by optimizing the upper-level master network. HRL was also applied in the energy management strategy, which solves the problem of the sparse reward and achieves higher training efficiency [33]. Although HRL can improve the efficiency of solving large-scale models and solve the problem of sparse rewards, it usually requires artificially setting goals to achieve good results, and there is no better way to improve this process.

## 3. Problem Formulation

In this section, we formulate the problem of crude oil supply chain scheduling. As shown in Figure 1, the system involves four facilities: oil fields, import ports, transfer stations, and refineries. The crude oil supply chain scheduling problem is a sequential decision-making problem. As crude oil is dynamically provided by oil fields and import ports at each time step and transported to transfer stations and refineries, the processing and transportation plans should be designed to satisfy system constraints such as the inventory and processing capacity. In order to facilitate the correspondence between symbols and definitions, Table 1 lists the notations defined throughout the paper.

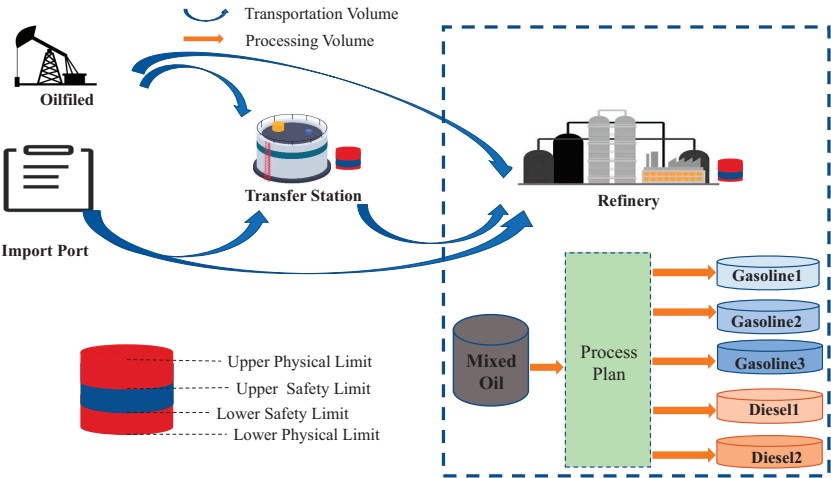

**Figure 1.** Crude oil supply chain scheduling problem.

**Table 1.** Notations.

| Symbols | Definitions |
|---|---|
| $M = (T, K, V, E, C, L)$ | A crude oil supply chain scheduling problem |
| $t, T$ | A time step and the number of total time steps |
| $k, K$ | A material kind and the set of material kinds |
| $c, d, g$ | The material kind of crude, diesel, and gasoline |
| $v, V, \mathbf{V}$ | A facility, facility set of a set, and facility set of all types |
| $e, E, \mathbf{E}$ | A road, road set related to a facility, and the set of all roads |
| $o, p, f, r$ | Oilfield, import port, transfer, and refinery |
| $C$ | The set of constraints |
| $LV = \{l_i^k \| v_i \in \mathbf{V}, k \in \mathbf{K}\}$ | The lower safety inventory limits |
| $UV = \{u_i^k \| v_i \in \mathbf{V}, k \in \mathbf{K}\}$ | The upper safety inventory limits |
| $PV = \{ph_i^k \| v_i \in \mathbf{V}, k \in \mathbf{K}\}$ | The physical inventory limits |
| $LQ = \{l_{q_r}^t \| v_r \in V_r, t = 1, \dots, T\}$ | The lower bound of processing volumes |
| $UQ = \{u_{q_r}^t \| v_r \in V_r, t = 1, \dots, T\}$ | The upper bound of processing volumes |
| $TQ = \{U_{q_r} \| v_r \in V_r\}$ | The upper bound of total processing volumes |
| $UE = \{u_{e_{ij}} \| e_{ij} \in \mathbf{E}\}$ | The upper limits of transportation volumes |
| $L$ | The set of constant coefficients |
| $\zeta = \{\zeta_i \| v_i \in \mathbf{V}\}$ | The unit cost of oil in facilities |
| $\lambda = \{\lambda_{e_{ij}} \in \mathbf{E}\}$ | The unit cost of oil on roads |
| $m, n$ | The inventory volume and the demand volume in facilities |
| $pr, \rho$ | The processing plan and the proportions in processing plan |
| $q$ | Decision of the processing volume plan in refineries |
| $w$ | Decision of the transportation volume on roads |
| $\mathcal{P}, \mathcal{H}, \mathcal{R}$ | The alert penalty, the transportation cost, and the total cost |
| $s, a, R$ | The state, action, and reward of a Markov decision process (MDP) |
| $Q$ | The Q-function in Q-learning algorithm |
| $or$ | The solving order |
| $J$ | The objective function |
| $x$ | The decision variables |
| $\alpha, \beta, \epsilon, \mu$ and $N$ | Hyperparameters |

The crude oil supply chain scheduling problem is formally defined as $M = (T, K, V, E, C, L)$, in which $T, K, V, E, C$, and $L$ represent the total time steps and the set of petroleum materials, facilities, roads, constraints, and constants, respectively. To be specific,

- The petroleum material set $K$ contains crude $c$, diesel $d$, and gasoline $g$. Let $k \in K$ represent one of the petroleum materials.

- Define the facility set $V = \{V_o, V_p, V_f, V_r\}$, where $V_o$ is the set of oil fields, $V_p$ is the set of import ports, $V_f$ is the set of transfer stations, and $V_r$ is the set of refineries. For convenience, let $V_i \in V$ be the set of facilities of type $i$, where $i = o, p, f, r$, and $v_i \in V_i$ is a single facility of type $i$. In each facility, $v_i$, the inventory volume of petroleum kind $k$ at time step $t$ is denoted as $m_i^{k,t}$, and the demand volume of the petroleum kind $k$ at time step $t$ is $n_i^{k,t}$. For each refinery, $v_r$, $q_r^t$ represents the processing volume of crude oil at time step $t$, and a fixed processing scheme $pr_r$ determines that the $\rho_r^d$ unit of diesel and $\rho_r^g$ unit of gasoline can be produced from 1 unit of crude oil.
- Define the set of roads $E = \{E_{of}, E_{or}, E_{pf}, E_{pr}, E_{fr}\}$, where $E_{ij} \in E$ is the set of transport roads from facility $v_i$ to facility $v_j$ (i.e., $j = o, p, f, r$ is the facility type). Each $e_{ij} \in E_{ij}$ represents one road from $v_i$ to $v_j$, and the set of roads starting from facility $v_i$ is denoted as $E_i$. In addition, the transportation volume on road $e_{ij}$ at time step $t$ is denoted as $w_{e_{ij}}^t$, and if there is no road from $v_i$ to $v_j$, $w_{e_{ij}}^t = 0$.
- The constraints of the crude oil supply chain scheduling problem are defined in $C$, which contains seven sets: $LV = \{l_i^k | v_i \in V, k \in K\}$ and $UV = \{u_i^k | v_i \in V, k \in K\}$ are the lower and upper safety inventory limits of facility $v_i$ for petroleum kind $k$, $PV = \{ph_i^k | v_i \in V, k \in K\}$ is the upper physical inventory limit of facility $v_i$ for the petroleum kind $k$, $LQ = \{l_{q_r}^t | v_r \in V_r, t = 1, \ldots, T\}$ and $UQ = \{u_{q_r}^t | v_r \in V_r, t=1,\ldots,T\}$ are the lower and upper bound of the processing volume in refinery $v_r$ at time step $t$, $TQ = \{U_{q_r} | v_r \in V_r\}$ is the upper bound of total processing volume in refinery $v_r$ for the whole $T$ steps, and $UE = \{u_{e_{ij}} | e_{ij} \in E\}$ is the upper limit of the transportation volume on road $e_{ij}$.
- The constant coefficients related to the system cost are denoted as $L = \{\zeta, \lambda\}$, where each $\zeta_i \in \zeta$ is the unit cost of facility $v_i$ when its inventory volume $m_i^{k,t}$ exceeds upper limit $u_i^k$ or lower safe limits $l_i^k$, and each $\lambda_{e_{ij}} \in \lambda$ is the unit cost of the road $e_{ij}$ as $w_{e_{ij}}^t \neq 0$.

At each time step $t$, the amount of crude oil from oil fields and import ports is randomly generated from the environment and is observed in real-time. Denote $w_{e_{of}}^t$, $w_{e_{or}}^t$, $w_{e_{pf}}^t$, and $w_{e_{pr}}^t$ as the supplied volume from oil field $v_o$ to transfer station $v_f$, oil field $v_o$ to refinery $v_r$, import port $v_p$ to transfer station $v_f$ and import port $v_p$ to refinery $v_r$. Meanwhile, the demand volume $n_i^{k,t}$ of each facility $v_i$ is observed from the environment. The processing volume plan $q_r^t$ and transportation plan $w_{e_{fr}}^t$ are controlled by our system, then the corresponding inventory volume $m_i^{k,t}$ of each facility can be deduced as follows:

$$m_f^{c,t} = m_f^{c,t-1} + \sum_{e_{of} \in E_{of}} w_{e_{of}}^t + \sum_{e_{pf} \in E_{pf}} w_{e_{pf}}^t$$
$$- \sum_{e_{fr} \in E_f} w_{e_{fr}}^t - n_f^{c,t} \tag{1}$$

$$m_r^{c,t} = m_r^{c,t-1} + \sum_{e_{or} \in E_{or}} w_{e_{or}}^t + \sum_{e_{pr} \in E_{pr}} w_{e_{pr}}^t - n_r^{c,t}$$
$$+ \sum_{e_{fr} \in E_f} w_{e_{fr}}^t - q_r^t \tag{2}$$

$$m_r^{d,t} = m_r^{d,t-1} + \rho_r^d q_r^t - n_r^{d,t} \tag{3}$$

$$m_r^{g,t} = m_r^{g,t-1} + \rho_r^g q_r^t - n_r^{g,t} \tag{4}$$

Equation (1): The transfer station $v_f$ receives crude oil $c$ from all oil fields to which the import ports connected with the amount of $w_{e_{of}}^t$ and $w_{e_{pf}}^t$ ($w_{e_{if}} = 0$ if $v_i$ and $v_f$ are disconnected) sends crude oil $c$ to all connected refineries with the amount $w_{e_{fr}}^t$ and provides a $n_f^{c,t}$ amount of crude oil $c$ on demand. Equation (2): Different from the transfer

station, a refinery $v_r$ also receives crude oil $c$ from all connected transfer stations with amount $w_{e_{fr}}^t$ and processes the amount $q_r^t$ of crude $c$ into gasoline and diesel. Equation (3): At time step $t$, the $\rho_r^d q_r^t$ amount of diesel is processed out, and the $n_r^{d,t}$ amount of it is deducted from the refinery on demand. Equation (4) has the same logic as Equation (3).

When the inventory volume exceeds safety limitations $LV$ and $UV$, the alert penalty at time step $t$ is

$$
\begin{aligned}
\mathcal{P}^t = & \sum_{v_i \in \boldsymbol{V}, k \in K} \zeta_i (m_i^{k,t} - u_i^k) \mathbb{1}[m_i^{k,t} > u_i^k] \\
& + \sum_{v_i \in \boldsymbol{V}, k \in K} \zeta_i (l_i^k - m_i^{k,t}) \mathbb{1}[m_i^{k,t} < l_i^k]
\end{aligned}
\tag{5}
$$

Moreover, the total transportation cost of the system at time step $t$ is

$$
\mathcal{H}^t = \sum_{e_{fr} \in E_{fr}} \lambda_{e_{fr}} w_{e_{fr}}^t
\tag{6}
$$

This problem aims to minimize the total cost of $T$ steps. At the same time, the system keeps all petroleum volumes under physical limitations (Equations (7b) and (7c)) and processing volumes under capacity limitations (Equations (7d) and (7e)):

$$
\max_{w_{e_{fr}}, q_r} J = \sum_{t=1}^T -\alpha_1 \mathcal{P}^t - \alpha_2 \mathcal{H}^t
\tag{7a}
$$

$$
\text{s.t.} \, 0 \le m_i^{k,t} \le ph_i^k \quad \forall v_i \in \boldsymbol{V}, k \in K, t = 1, \dots, T
\tag{7b}
$$

$$
0 \le w_{e_{ij}}^t \le u_{e_{ij}} \quad \forall e_{ij} \in \boldsymbol{E}, t = 1, \dots, T
\tag{7c}
$$

$$
l_{q_r}^t \le q_r^t \le u_{q_r}^t \quad \forall v_r \in V_r, t = 1, \dots, T
\tag{7d}
$$

$$
\sum_{t=1}^T q_r^t = U_{q_r} \quad \forall v_r \in V_r
\tag{7e}
$$

where $\alpha_1$ and $\alpha_2$ are weights.

The difficulties of this problem lay in two aspects: (1) Large problem scale. We aim to solve the crude oil supply chain scheduling problem in the real world, which includes hundreds of facilities and thousands of roads, and the decision period lasts about 30-time steps. These form a decision space of about $10^7$ with continuity. (2) Uncertain environment. Environment variables are not deterministic but are generated in real-time, which means the variables at time step $t + 1$ cannot be observed at time step $t$. Suppose the environment variables of $T$ steps are known in advance; in that case, the problem can be formulated as a Linear Programming (LP) problem which can be solved by powerful solvers (e.g., Gurobi [34]). However, this kind of method cannot be applied to our problem directly due to uncertainty. On the other hand, as the one-step problem is also an LP problem, an intuitive way might be solving the multi-step problem step by step. However, traditional solvers are unable to take the future influence of the current solution into consideration, thus leading to sub-optimal solutions when dealing with multi-step problems, and even failing to find a feasible solution after several steps (as shown in Section 6).

## 4. Hierarchical RL Scheduling Scheme

In this section, we demonstrate the whole architecture of our framework, including the upper-level and lower-level agents. Necessary constraints are also described in this section.

### 4.1. Overall Architecture

As described in Section 2, the main challenges of this problem are the large decision space and uncertainty. Traditional solvers are powerful tools to solve large-scale LP problems but cannot deal with uncertainty. In contrast, RL methods can solve problems with

stochastic state transitions but are hard to train when the problem scale is enormous. Considering the property of this problem, we design an HRL-based optimization architecture, as shown in Figure 2 and Algorithm 1.

---

**Algorithm 1** Hierarchical RL Scheduling Algorithm

---
1: Initialize model parameters $\theta$
2: **while** Not converged **do**
3:　　Observe initial state $s_0$
4:　　Initialize $\tilde{U}_{q_r}^0 = U_{q_r}$
5:　　**for** t=1 to $T$ **do**
6:　　　　Select action $a_t \sim Q_\theta(s_t, a_t)$ by Section 4.2.2
7:　　　　Solve the LP problem Equation (18a), where $m_i^k \leftarrow a_t$, $U_{q_r} \leftarrow \tilde{U}_{q_r}^t$, get $w_{e_{fr}}^t$ and $q_r^t$
8:　　　　Transit from $s_t$ to $s_{t+1}$ by Equations (11)–(16)
9:　　　　Set $\tilde{U}_{q_t}^t \leftarrow \tilde{U}_{q_r}^{t-1} - q_r^t$
10:　　**end for**
11: **end while**

---

The entire crude oil supply chain scheduling problem involves $T$ steps. The supply volume of crude oil and the demand volume for diesel and gasoline at each step is dynamic, forming a stochastic environment. Moreover, the transportation plan and processing volume plan decided in each step will affect future decisions. The impacts are reflected explicitly in two ways: the initial inventory of the future steps and the remaining available processing volume, since there is a constraint on total processing volumes $U_{q_r}$. The latter can be dealt with shaped constraints at each time step with $\tilde{U}_{q_r}^t \leftarrow \tilde{U}_{q_r}^{t-1} - q_r^t$ and $\tilde{U}_{q_r}^1 = U_{q_r}$. Then, suppose the initial inventory $m_i^{k,t}$, the ending inventory $m_i^{k,t+1}$, the supply volume $w_{e_{of}}^t$, $w_{e_{or}}^t$, $m_{e_{pf}}^t$ and $w_{e_{pr}}^t$, and the demand volume $n_i^{k,t}$ at time step $t$ are given; then, the problem for time step $t$ is an LP problem, independent from other time steps. Therefore, we can use an RL agent to learn the inventory volume and use a solver to solve the LP problem.

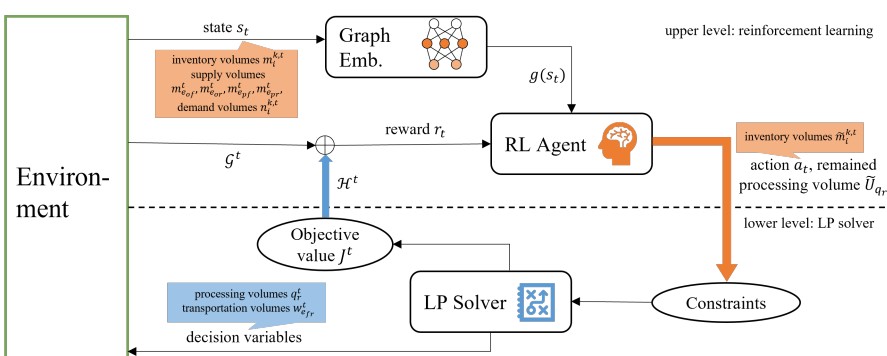

**Figure 2.** Overall Architecture.

The upper layer is the RL layer. The RL agent takes the inventory volumes of each facility at the end of the time step $t$ as the action, denoted as $\tilde{m}_i^k$. The goal of the RL agent is to minimize the summation of the alert penalty for multiple steps and the transportation penalty, as defined in Equation (7a). However, using the hierarchical framework, the transportation penalty is received from the lower-layer agent.

The lower layer is a traditional optimizer layer, which solves the LP problem Equation (7a) of one time step. At time step $t$, the solver receives the end-stage inventory volumes $\tilde{m}_i^k$, and the remaining processing volume of each refinery $q_r^t$ from the upper-level agent uses them to form the one-step constraints which split from the multi-steps constraints in Equations (7b) and (7e). The lower layer takes the process volume plan $q_r^t$ and the transportation plan $w_{e_{fr}}^t$ of the time step $t$ as decision variables and minimizes the total transportation costs $\mathcal{H}^t$ defined in Equation (6), which is passed back to the upper-level agent.

In addition, based on experience, we know that RL performs better on small-scale discrete problems; thus, we propose a variant for our crude oil supply chain scheduling problem. We discretize the action space into several end-stage inventory values and add one dim, indicating the solving orders of the solver. Compared to the vanilla architecture, the lower layer of the variant architecture receives an extra signal, determining the solving order of $w_{e_{fr}}^t$ and $q_r^t$.

### 4.2. Upper-Level Agent

The decision interval of the upper agent is one time step, and there are $T$ steps in each episode. For $t = 1, \ldots, T$, the environment stochastically provides the supply volume of crude oil from each oil field to each transfer station or refinery $w_{e_{of}}^t$ and $w_{e_{or}}^t$, the supply volume from each import port to each transfer station or refinery $w_{e_{pf}}^t$ and $w_{e_{pr}}^t$, and the demand volume for gasoline or diesel in the transfer station and refinery $n_i^{k,t}$. The RL agent observes status $s_t$ from the environment and determines the expected inventory volume of each refinery and transfer station at the end of this time step $t$, noted as $a_t$. The upper-level agent will provide these end-stage inventory volumes to the lower-level agent, which will search out an actual execution strategy. The environment moves to the next time step $t + 1$, based on the actual strategy. The reward $R_t$ is calculated by combining the inventory alert penalty $\mathcal{P}^t$ and the feedback $J^t$ from the lower-level agent.

It is worth noting that at the beginning of training, the lower-level agent may not find feasible solutions if it intends to strictly meet the expected inventory $a_t$. Therefore, at the lower layer, these expected inventory values are not used as hard constraints (see Section 4.3 for specification), and the inventory volumes caused by the execution strategy (including processing volume and transportation plan) are not necessarily equal to the expected inventory. In other words, from the perspective of the upper-level agent, although the action is the expected inventory at the end of the time step $t$, i.e., the expected inventory at the beginning of the next time step, the actual observed inventory from the environment at time step $t + 1$ is not necessarily equal to the action $a_t$. In the next part, we provide the Markov decision processes (MDP) corresponding to the description.

#### 4.2.1. Markov Decision Process (MDP)

The state of the RL agent includes all kinds of variable attributes in the facilities related to the problem, denoted as $s_t = \{w_{e_{of}}^t, w_{e_{or}}^t, w_{e_{pf}}^t, w_{e_{pr}}^t, m_f^{c,t}, m_r^{c,t}, m_r^{d,t}, m_r^{g,t}, \tilde{U}_{q_r}^t\}$, as detailed in Table 2.

**Table 2.** States of upper-level MDP.

| Note | Dim | Notations |
|------|-----|-----------|
| $w_{e_{of}}^t$ | $|E_{of}|$ | Crude supply volume from $v_o$ to $v_f$ |
| $w_{e_{or}}^t$ | $|E_{or}|$ | Crude supply volume from $v_o$ to $v_r$ |
| $w_{e_{pf}}^t$ | $|E_{pf}|$ | Crude purchase volume from $v_p$ to $v_f$ |
| $w_{e_{pr}}^t$ | $|E_{pr}|$ | Crude purchase volume from $v_p$ to $v_r$ |
| $m_f^{c,t}$ | $|V_f|$ | Crude inventory volume of $v_f$ |
| $m_r^{c,t}$ | $|V_r|$ | Crude inventory volume of $v_r$ |
| $m_r^{d,t}$ | $|V_r|$ | Diesel inventory volume of $v_r$ |
| $m_r^{g,t}$ | $|V_r|$ | Gasoline inventory volume of $v_r$ |
| $\tilde{U}_{q_r}^t$ | $|V_r|$ | Remained processing volume of $v_r$ |

The actions of time step $t$ of the upper-level agent are expected inventory volumes of crude, gasoline, and diesel in transfer stations and refineries, denoted as $a_t = \{\tilde{m}_f^{c,t+1}, \tilde{m}_r^{c,t+1},$

$\tilde{m}_r^{d,t+1}, \tilde{m}_r^{g,t+1}\}$, and the total $|V_f| + 3 \times |V_r|$ dimensions. Each dimension is continuous and bounded by the physical inventory limit $ph_i^k$.

The reward function Equation (8) consists of two parts. One is the inventory alert penalty $\mathcal{P}^t$, which can be calculated by substituting $a_t$ into Equation (5). The other is the feedback penalty $J^t$ returned by the lower-level agent. The detailed definition of the feedback penalty $J^t$ is described in Equation (18a); conceptually, it includes the total transportation cost $\mathcal{H}^t$, as defined in Equation (6) and the deviation of $\tilde{m}_i^k$ from constraints in Equation (7b). This encourages the actions of the upper-level agent to optimize the objective of the crude oil supply chain scheduling problem as well as satisfies the constraints.

$$R(s_t, a_t) = -\alpha_1 \mathcal{P}^t(a_t) - \alpha_2 J^t \tag{8}$$

where $\alpha_1$ and $\alpha_2$ are the same parameters defined in Equation (7a).

### 4.2.2. Agent Model

We use the Q-learning algorithm for the upper-level agent, a model-free, on-policy algorithm. We parameterize a Q-network $Q_\theta$ and update it as follows.

$$Q_\theta(s_t, a_t) \leftarrow R(s_t, a_t) + \gamma \max Q_\theta(s_{t+1}, a_{t+1}) \tag{9}$$

where $\gamma$ is a discount factor.

The action is selected according to $Q_\theta$ using the $\epsilon-$greedy policy:

$$a_t = \begin{cases} \arg\max_a Q_\theta(s_t, a_t) & w.p. \quad 1 - \epsilon \\ random & w.p. \quad \epsilon \end{cases} \tag{10}$$

### 4.3. Lower-Level Agent

The goal of the lower-level agent is to search for a feasible execution strategy (including a processing volume $q_r^t (\forall v_r \in V_r)$ and a transportation plan $w_{e_{fr}}^t (\forall e_{fr} \in E_{fr})$) in the current time step $t$, which minimizes the overall transportation cost $\mathcal{H}^t$ and satisfies the one-step constraints. In other words, the lower-level agent solves the one-step problem of Equation (7a). Constraints are composed of two parts; one is obtained from the upper agent, including $a_t$ representing the end-stage inventory volume of each facility to substitute Equation (7b) and $\tilde{U}_{qr}$ representing the remained processing volume until time step $t$ to substitute Equation (7e), and the other part is obtained according to the problem itself, including processing constraints in refineries as Equation (7d) and the transportation volume constraints on the roads as Equation (7c).

Since the constraints and the objective function are linear concerning the processing volume $q_r^t$ and transportation volume $w_{e_{fr}}^t$, this problem can be formalized as a large LP problem and solved with traditional solvers. However, because $a_t$ might be unreasonable at the beginning of training, without dealing with the constraints, this LP problem is likely to be unsolvable. In this case, as a lower-level agent, the emergence of no solution not only makes the decision-making of this step fail but also terminates this training episode. In consequence, the upper-level agent is held back on the convergence speed.

To solve this problem, we relax the constraints. The main idea is to relax all equality constraints, design corresponding barrier functions according to each original equality condition, and add them to the objective function. In this way, in most cases, the linear programming problem has a solution, and the objective function value $J$ corresponding to the optimal solution reflects the quality of $a_t$, which can be sent back to the upper agent as a suitable incentive signal.

4.3.1. LP Formulation

We define decision variables $x_{e_{fr}}$ as a transportation volume of crude on the road $e_{fr}$ from transfer station $v_f$ to refinery $v_r$ at time step $t$ (i.e., $w_{e_{fr}}^t$), and $x_r$ as a processing volume of crude oil in refinery $v_r$ at time step $t$ (i.e., $q_r^t$), respectively.

Since the LP problem is independent of time step $t$, we omit the superscript $t$ in the LP formulation for simplicity. We rewrite the definition Equations (1)–(4) of the inventory volumes here, and we call $m_f^c$, $m_r^c$, $m_r^d$ and $m_r^g$ the auxiliary variables.

$$m_f^c \leftarrow m_f^c + \sum_{e_{of} \in E_{of}} w_{e_{of}} + \sum_{e_{pf} \in E_{pf}} w_{e_{pf}} \tag{11}$$

$$- \sum_{e_{fr} \in E_{fr}} x_{e_{fr}} - n_f^c \tag{12}$$

$$m_r^c \leftarrow m_r^c + \sum_{e_{or} \in E_{or}} w_{e_{or}} + \sum_{e_{pr} \in E_{pr}} w_{e_{pr}} + \sum_{e_{fr} \in E_{fr}} x_{e_{fr}}^c \tag{13}$$

$$- n_r^c - x_r \tag{14}$$

$$m_r^d \leftarrow m_r^d + \rho_r^d x_r^c - n_r^d \tag{15}$$

$$m_r^g \leftarrow m_r^g + \rho_r^g x_r^c - n_r^g \tag{16}$$

Then, the original LP problem can be formulated as:

$$\min_{x_{e_{fr}}, x_r} J = \mathcal{H}^t = \sum_{e_{fr} \in E_{fr}} \lambda_{e_{fr}} x_{e_{fr}} \tag{17a}$$

$$\text{s.t.} \quad m_f^c = \tilde{m}_f^c, \quad \forall v_f \in V_f \tag{17b}$$

$$m_r^c = \tilde{m}_r^c, \quad \forall v_r \in V_r \tag{17c}$$

$$m_r^d = \tilde{m}_r^d, \quad \forall v_r \in V_r \tag{17d}$$

$$m_r^g = \tilde{m}_r^g, \quad \forall v_r \in V_r \tag{17e}$$

$$x_r \le \tilde{U}_{q_r}, \quad \forall v_r \in V_r \tag{17f}$$

$$l_{q_r} \le x_r \le u_{q_r}, \quad \forall v_r \in V_r \tag{17g}$$

$$0 \le x_{e_{fr}} \le u_{e_{fr}}, \quad \forall e_{fr} \in E_{fr} \tag{17h}$$

The objective is to minimize the total transportation costs $\mathcal{H}^t$, as shown in Equation (17a) (equivalent to Equation (6)) when meeting conditions of inventory capacity in Equations (17b) and (17e) (split from Equation (7b)), processing capacity in Equation (17f) and (17g) (split from Equations (7e) and (7d)), and transportation capacity in Equation (17h) (split form Equation (7c)). According to the overall architecture, the auxiliary variables $m_i^k$ representing end-stage inventory volumes are required not to deviate from the inventory suggestions of the upper agent.

To guarantee that a feasible solution can be searched for in any $a_t$, so that the lower-level agent always passes back a meaningful value $J$ to the upper-level agent, we relax the constraints and reformulate the LP problem with barrier functions $B(m)$, which are detailed as described in Section 4.3.2.

$$\min_{x_{e_{fr}}, x_r} J = \mu_1 \mathcal{H}^t - \mu_2 B(m_f^c) - \mu_3 B(m_r^c) -$$

$$\mu_4 B(m_r^d) - \mu_5 B(m_r^g) \tag{18a}$$

$$\text{s.t.} \quad x_r \le U_{q_r}, \quad \forall v_r \in V_r \tag{18b}$$

$$l_{q_r} \le x_r \le u_{q_r}, \quad \forall v_r \in V_r \tag{18c}$$

$$0 \le x_{e_{fr}} \le u_{e_{fr}}, \quad \forall e_{fr} \in E_{fr} \tag{18d}$$

where $\mu_1, \dots, \mu_5$ are coefficients of the different parts of the objective function.

#### 4.3.2. Barrier Function

The purpose of designing the barrier function is to convert the hard constraints on the auxiliary variables into penalty items in the objective function. Each auxiliary variable $m_i^k$ representing the end-stage inventory volume of the material kind $k$ in facility $v_i$ corresponds to an expected inventory volume $\tilde{m}_i^k$ given by the upper-level agent. We hope that $m_i^k$ is as close to $\tilde{m}_i^k$ as possible. The greater the deviation from $\tilde{m}_i^k$, the greater the penalty. In addition, the problem requires that the storage $m_i^k$ is within the physical inventory limit $ph_i^k$ all the time, and preferably within the upper and lower safety inventory limit of $u_i^k$ and $l_i^k$; then, for $m_i^k$, which deviates a lot from $\tilde{m}_i^k$ and thus exceeds the upper and lower safety inventory limit or the physical inventory limit, a higher penalty should be given.

Based on these considerations, we design the barrier function as a piece-wise linear function, as shown in Figure 3. The barrier function of $m_i^k$ is expressed mathematically as Equation (19), where $\beta_1 < \beta_2 < \beta_3$, $\Delta_1 = u_i^k - l_i^k$, $\Delta_2 = l_i^k - 0$, $\Delta_3 = ph_i^k - u_i^k$.

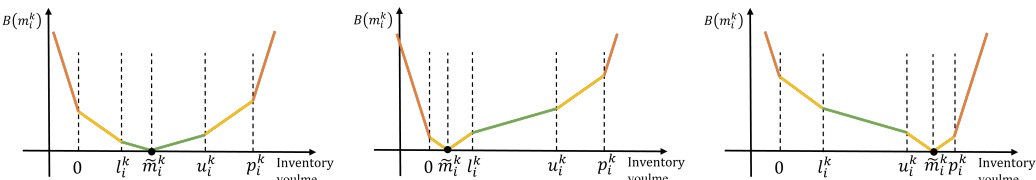

**Figure 3.** Barrier function: (**Left**) $l_i^k < \tilde{m}_i^k \le u_i^k$. (**Middle**) $0 < \tilde{m}_i^k \le l_i^k$. (**Right**) $u_i^k < \tilde{m}_i^k \le ph_i^k$.

$$
B(m_i^k) = \begin{cases}
\begin{cases}
\beta_1|l_i^k - \tilde{m}_i^k| + \beta_2\Delta_2 + \beta_3|m_i^k - 0| & \text{if } m_i^k < 0 \\
\beta_1|l_i^k - \tilde{m}_i^k| + \beta_2|m_i^k - l_i^k| & \text{if } 0 < m_i^k \le l_i^k \\
\beta_1|m_i^k - \tilde{m}_i^k| & \text{if } l_i^k < m_i^k \le u_i^k \\
\beta_1|u_i^k - \tilde{m}_i^k| + \beta_2|m_i^k - u_i^k| & \text{if } u_i^k < m_i^k \le ph_i^k \\
\beta_1|u_i^k - \tilde{m}_i^k| + \beta_2\Delta_3 + \beta_3|m_i^k - ph_i^k| & \text{if } ph_i^k < m_i^k
\end{cases} & \text{if } l_i^k < \tilde{m}_i^k \le u_i^k \\[4em]
\begin{cases}
\beta_2|0 - \tilde{m}_i^k| + \beta_3|m_i^k - 0| & \text{if } m_i^k < 0 \\
\beta_2|m_i^k - \tilde{m}_i^k| & \text{if } 0 < m_i^k \le l_i^k \\
\beta_2|l_i^k - \tilde{m}_i^k| + \beta_1|m_i^k - \tilde{m}_i^k| & \text{if } l_i^k < m_i^k \le u_i^k \\
\beta_2|l_i^k - \tilde{m}_i^k| + \beta_1\Delta_1 + \beta_2|m_i^k - u_i^k| & \text{if } u_i^k < m_i^k \le ph_i^k \\
\beta_2|l_i^k - \tilde{m}_i^k| + \beta_1\Delta_1 + \beta_2\Delta_3 + \beta_3|m_i^k - ph_i^k| & \text{if } ph_i^k < m_i^k
\end{cases} & \text{if } 0 < \tilde{m}_i^k \le l_i^k \\[4em]
\begin{cases}
\beta_2|u_i^k - \tilde{m}_i^k| + \beta_1\Delta_1 + \beta_2\Delta_2 + \beta_3|m_i^k - 0| & \text{if } m_i^k < 0 \\
\beta_2|u_i^k - \tilde{m}_i^k| + \beta_1\Delta_1 + \beta_2|m_i^k - l_i^k| & \text{if } 0 < m_i^k \le l_i^k \\
\beta_2|u_i^k - \tilde{m}_i^k| + \beta_1|m_i^k - u_i^k| & \text{if } l_i^k < m_i^k \le u_i^k \\
\beta_2|m_i^k - \tilde{m}_i^k| & \text{if } u_i^k < m_i^k \le ph_i^k \\
\beta_2|ph_i^k - \tilde{m}_i^k| + \beta_3|m_i^k - ph_i^k| & \text{if } ph_i^k < m_i^k
\end{cases} & \text{if } u_i^k < \tilde{m}_i^k \le ph_i^k
\end{cases}
\tag{19}
$$

#### 4.4. The Variant on Discrete Action Space

Exploring the optimal policy in a discrete action space with low dimensions for an RL agent is much easier and more stable than in a continuous action space. Moreover, we aim to solve the crude oil supply chain scheduling problem in the real world, which requires the algorithm to converge quickly in a stable manner. Therefore, we design a variant that combines 9 operators as the action space. The operators are shown in Table 3. The action of the RL agent is to select one operator for each corresponding node and one operator for the solving order $or^t$, forming an action space of $|V_r| + |V_r| + |V_f| + 1$ dimensions.

**Table 3.** Operators of the variant algorithm.

| Diesel and Gasoline in Refineries | Crude in Refineries | Crude in Transfer Stations | Solving Order |
|---|---|---|---|
| Upper safety limit | Periodic demand | Increase by 10% | Simultaneously |
| Lower safety limit | Upper safety limit | Decrease by 10% | Sequentially |
| Current inventory | - | - | - |

The crude oil supply chain receives crude oil from oil fields and import ports and provides diesel and gasoline on demand. Then, the inventory of each node can be stable if the production and demand are balanced. In other words, the end-stage inventory of nodes is a baseline inventory that needs to be dynamically adjusted in case the supply and demand are unbalanced.

The demand for diesel and gasoline is uncertain at each time step; thus, we designed 3 naive operators:

- To meet the upper safety limit, which means $\tilde{m}_r^{d,t} = u_r^d$ and $\tilde{m}_r^{g,t} = u_r^g$;
- To meet the lower safety limit, which means $\tilde{m}_r^{d,t} = l_r^d$ and $\tilde{m}_r^{g,t} = l_r^g$;
- To maintain the current inventory, which means $\tilde{m}_r^{d,t} = m_r^{d,t-1}$ and $\tilde{m}_r^{g,t} = m_r^{g,t-1}$.

Since diesel and gasoline are produced proportionally, to control the crude inventory in refineries, we approximate the demand for crude oil accordingly:

$$\tilde{n}_r^{c,t} = \max\left\{ \frac{n_r^{d,t}}{\rho_r^d}, \frac{n_r^{g,t}}{\rho_r^g} \right\} \tag{20}$$

The refinery could process crude oil into diesel and gasoline if its crude oil inventory is about to overflow, but it is hard to replenish the inventory when the crude oil supply is short; thus, the operators for the end-stage inventory of crude oil are designed as follows:

- To meet periodic demand, which means $\tilde{m}_r^{c,t} = N \cdot \tilde{n}_r^{c,t}$, where $N$ is a length of time step period;
- To meet the upper safety limit, which means $\tilde{m}_r^{c,t} = u_r^c$.

To avoid the sharp rise and fall of the inventory, the operators of transfer stations are designed as follows:

- To increase by 10%, which means $\tilde{m}_f^{c,t} = 1.1 \times m_f^{c,t-1}$;
- To decrease by 10%, which means $\tilde{m}_f^{c,t} = 0.9 \times m_f^{c,t-1}$.

In addition, we use another operator to control the solving order of the lower-level agent. The idea is that when the processing volumes $q_r^t$ in refineries are settled, the amount of crude oil consumed at this time step is known, and the approximate range of the amount of crude oil that the refineries need to receive from the transfer stations can be deduced, facilitating the optimization of $w_{e_{fr}}^t$. On the contrary, if the transportation volumes $w_{e_{fr}}^t$ are settled first, there is a high probability that the refineries will not be able to meet the inventory requirements only by adjusting the processing volume of $q_r^t$. However, mindlessly optimizing $q_r^t$ first may cause inventory overflow of many transfer stations; thus, we use an operator $or^t$ to control the optimization order:

- To solve $q_r^t$ and $w_{e_{fr}}^t$ simultaneously, which means $or^t = 0$;
- To solve $q_r^t$ and $w_{e_{fr}}^t$ sequentially, which means $or^t = 1$.

At each time step $t$, the upper-level agent makes decisions in two steps: firstly, the RL agent selects one way of the combination of these operators, and secondly, $\tilde{m}_i^k$ and $or^t$ are calculated according to the operators. Then, the lower-level agent will solve the *LP* problem as discussed in Section 4.3 according to the solving order $or^t$.

## 5. Simulator

We create a simulator to evaluate and train RL algorithms to solve the problem described in Section 4. This section will introduce the simulator we used and the function of the simulator in training and testing.

### 5.1. Data for Simulator

During our research, we harnessed real-world data from an oil distribution company that produces and transports crude oil and refined products. This comprehensive dataset includes attributes associated with transportation modes and four types of nodes, namely: oil fields, import ports, transfer stations, and refineries, as stated in Section 3. Specifically, it encompasses refined product categories, inventory capabilities of refineries, sales stations, transfer nodes, and the demand figures from the aforementioned sales stations. In addition, our data also incorporate essential operational aspects such as transportation costs between nodes and conversion ratios from crude oil to refined products.

Certain preprocessing measures were employed for our experimental framework to address inherent anomalies in the raw dataset. Notably, missing data, denoted as 'NAN' in the raw dataset, were substituted based on contextual relevance; values of 0 and 999 were designated as lower and upper bounds, respectively. These preprocessed values were then assigned to their corresponding node classes to enhance the realism and robustness of our simulator. This simulator faithfully represents the intricate dynamics of a crude oil supply chain and serves as an integral part of our RL framework. For comparative experiments, we consider 26 supply nodes, 20 transfer nodes, 26 refineries, and 164 roads between each node to meet the daily demand for products and the inventory constraints. The dynamic simulator allows our model to interact with a continuously evolving environment, learn from it, and evaluate its performance.

### 5.2. Simulator Design

The simulator preprocesses the raw data from facilities and roads and provides continuous feedback to the algorithm in training and testing. The simulator also serves as an evaluation environment after training. In addition, our simulator offers essential indicators such as alert penalty and transportation cost that were mentioned in Section 3. Additionally, we also record the number of alerts triggered during the simulation. By analyzing these metrics, we can evaluate the performance of different algorithms in a systematic and quantitative manner.

## 6. Experiment

The experiments consist of: (1) using the same dataset to implement our method HRL and three other baselines, illustrating the average train loss and average test reward of HRL; (2) comparing three probable effect factors, including the alert number, the maximum alert penalties, and the transportation cost.

### 6.1. Experiment Settings

We compare four algorithms: the traditional OR linear programming solved by Gurobi, proximal policy optimization (PPO), soft actor–critic (SAC), and our method, HRL.

All of the baseline models use the same dataset and have 30-time steps per episode. For SAC and PPO, the action space and state space are 298 and 131, respectively, based on the simulator described in Section 5.1. For our method HRL, we set the replay buffer size for Q-network to 2,000,000 and the batch size to 1024. We use $\epsilon$-greedy for exploration during learning. The networks are trained with Adam [35] with a learning rate of $1 \times 10^{-5}$ and discount over training epochs. We implement the pipelines using Pytorch and train them on GPUs with Nvidia Geforce 3090 Ti. We average the results over 16 repeat runs. The average training loss and average testing reward of HRL are shown in Figures 4 and 5.

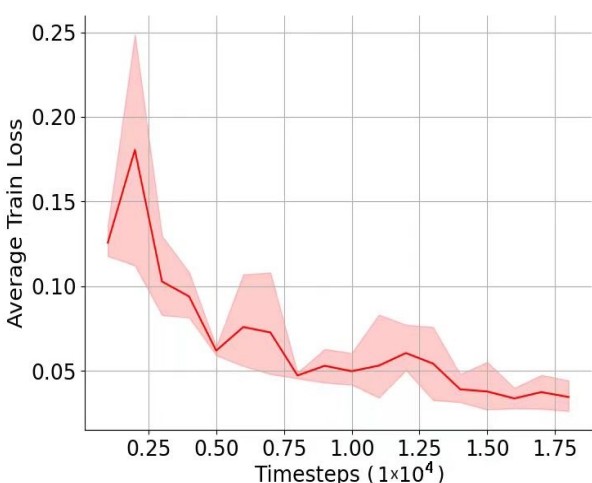

**Figure 4.** Average training loss of HRL.

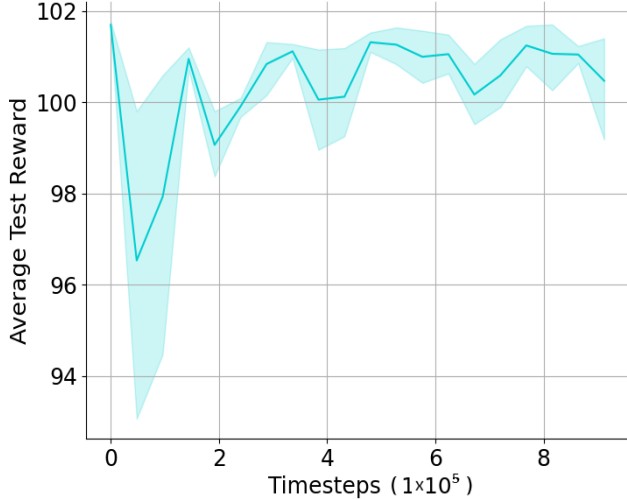

**Figure 5.** Average test reward of HRL.

### 6.2. The Comparison against Baselines

Although our goal is to avoid the frequency of inventory overrun alerts as much as possible, such alerts may inevitably occur due to limitations of the initial state of the environment determined by real-world data. We first compare the number of alerts and the maximum alert penalty to verify that our two-level HRL agents learn to make suitable scheduling plans at different time steps. The x-axis shows the time step in both Figures 6 and 7. The y-axis shows the exact number of alerts in each time step in Figure 6. OR and HRL perform better than traditional RL algorithms in most time steps. Overall, HRL generates fewer alerts than the other three algorithms. In Figure 7, we record the maximum alert penalty for 30-time steps. As a result of SAC and PPO, a gradual increase is shown by the curve in Figure 7. We think that the reason for the poor performance of these two RL algorithms may be that the action space and state space are too large. As mentioned in Section 6, such a large action space and state space for RL will make it difficult to train and obtain acceptable results.

Moreover, we compare the maximum alert penalty in percentage in Figure 7. After all, we mainly focus on those nodes with a low upper inventory limit, which is more important than the same alert penalty amount with a high upper limit because it increases the risk of exceeding the inventory limit. Similarly, OR and HRL results are much better than traditional RL algorithms.

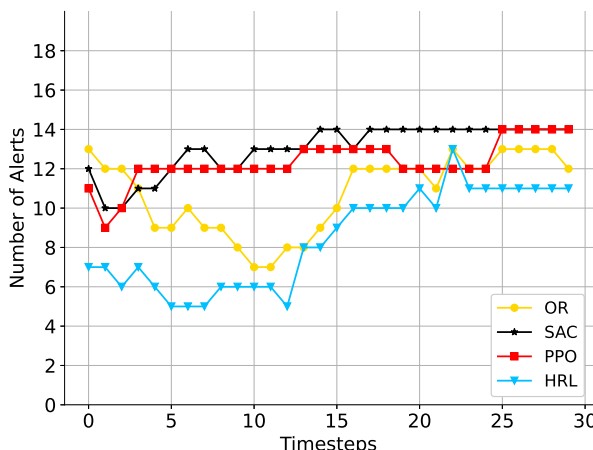

**Figure 6.** A comparison of the alert number at each time step.

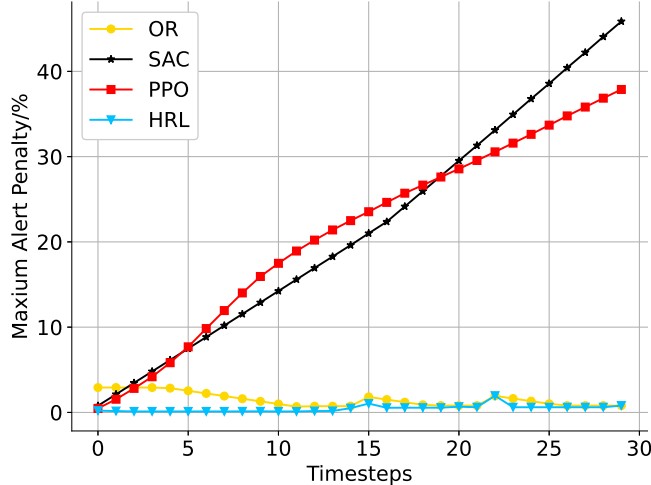

**Figure 7.** A comparison of maximum alert penalties at each time step in percentage.

Besides the alert penalty, transportation costs also matter for the objective function and algorithm effectiveness. In Figure 8, HRL generates transportation costs below 20,000 except for time step 15. The results of SAC and PPO are consistently higher than 20,000, and OR accomplishes this for approximately half of the time steps and fails to maintain costs at a consistently low level due to the transportation duration spanning multiple days; the scheduling of transportation volume cannot effectively align with changes in demand.

Another critical issue is the solving time of the algorithm, which holds great significance in balancing solution quality and computational costs. Table 4 demonstrates a comparative analysis of computational time, revealing that our approach exhibits the longest solution time among the considered methods. Nevertheless, it is noteworthy that our method remains considerably affordable, with only a slight deviation compared to other RL algorithms and its exceptional performance.

**Table 4.** A comparison of solution time.

| Algorithm | Solution Time (in Seconds) |
| --- | --- |
| OR | 0.58 |
| PPO | 0.71 |
| SAC | 0.68 |
| HRL | 1.17 |

In conclusion, our HRL can achieve the smallest warning penalties and the lowest overall transportation costs compared to other algorithms. Despite the higher computational cost, the prolonged duration enables the discovery of an optimal solution for the crude oil supply chain problem through its two-level design.

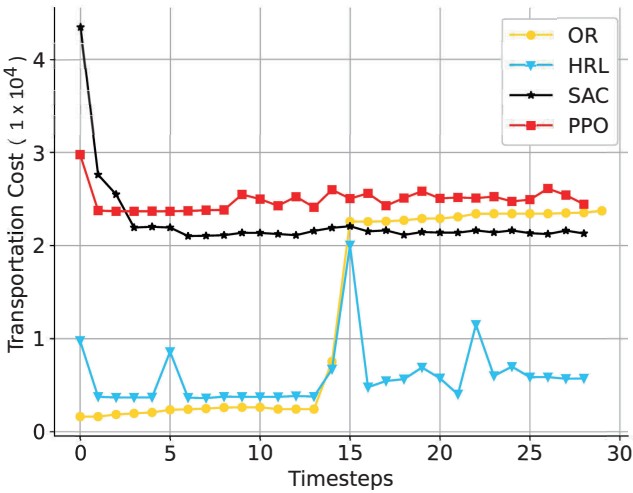

**Figure 8.** A comparison of transportation cost at each time step.

### 6.3. The Influence of Parameter N

As mentioned in Section 4.4, our HRL algorithm is designed to meet the target of $N$ time steps' demand (i.e., $N$ times "crude oil demand value"). We then compare the performance when $N$ equals 3, 5, and 7 steps in Figure 9. A suitable $N$ value leads to a balanced node distribution, resulting in relatively low alert penalties and transportation costs. We compare them to two aspects to demonstrate this parameter's effect on the results. The left y-axis in the figure shows the total transportation costs, and the right y-axis shows the total alert penalties over 30-time steps.

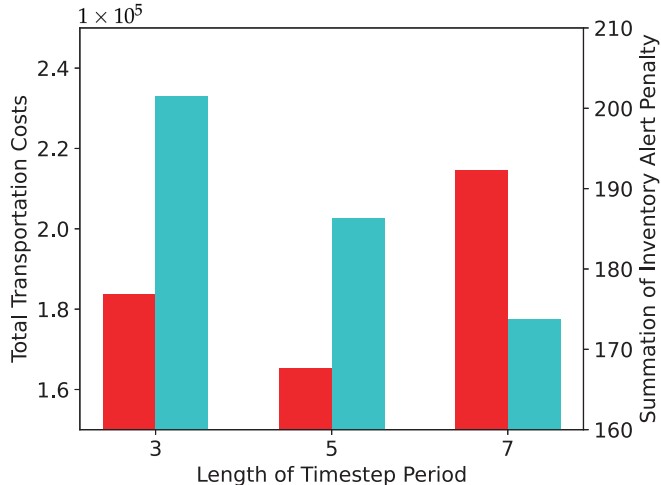

**Figure 9.** A comparison of HRL using different operator parameter to meet periodic demand.

### 6.4. Discussion

Compared to prior studies of supply chain scheduling or resource allocation issues, our hierarchical optimization framework demonstrated superior performance in considering both solution quality and transportation costs. Our methods align with research conducted by [11] and [25], who also modeled the scheduling problem in linear programming or the MDP form. However, our HRL algorithm surpasses the existing methodology by expanding

the planning horizon to more time steps and decomposing the large-scale problem with well-designed operations operators. One of the limitations of our proposed model is that the training and solving time is longer compared to RL algorithms such as PPO and SAC due to the increased complexity of the hierarchical framework. Additionally, taking the refinery node or transfer node as an example, selecting the same operator for all nodes implies that they all have the same optimization objective, resulting in a certain loss of flexibility. Consequently, it may not achieve the overall optimality for more complex problems.

## 7. Conclusions and Future Works

This paper presents a framework based on HRL to address the complex problem of petroleum transportation and processing. The upper-level agent is equipped with the foresight ability, whose target is to reduce the penalty caused by inventory limit violations, demand dissatisfaction, and infeasible plans. The lower-level agent makes decisions on the specific process strategy and transportation plan for the time step according to the end-stage inventory volume from the upper-level agent. It is worth noting that the combination of upper-level and lower-level agents allows our method to find globally better solutions that consider the effects on subsequent time steps. Online testing on a simulator based on real-world data demonstrated that our approach achieves better solutions than baselines. Moreover, our work is only a first step towards using HRL for larger-scale problems in real-world scenarios. Future research should be devoted to the development of the following parts.

- Problem Representation via Graph Neural Networks
  In future research, more research is needed to apply and test Graph Neural Networks (GNNs). GNNs can inherently work with graph-structured data. They utilize the graph structure and apply neural networks to learn from the graph data. Employing GNNs to represent nodes and edges interconnecting nodes in large-scale scheduling problems enables capturing node dependencies and relationships.
- Multi-agent Reinforcement Learning
  Multi-agent reinforcement learning algorithms can accomplish objectives through cooperation between agents. Concretely, each facility can be modeled as an individual agent. Each agent selects the optimal action so that the overall action is optimal. The flexibility of the framework can be significantly improved with the insurance of optimality. It will be important that future research explore how multi-agent algorithms can be designed, which nodes should be modeled as agents, and how credit can be assigned among these agents.
- Generalized Simulator and Operator Design
  While our simulator is based on real-world data and can handle large-scale scheduling problems, if there are new scheduling or other graph structure optimization problems, the simulator needs to be redesigned or reconstructed in many places. This is an issue for future research to explore. A more generalized simulation environment and operators have great potential for HRL to solve optimization problems.

**Author Contributions:** All authors contributed to the study conception and design. Conceptualization, investigation, data analysis, and model verification: N.M.; Writing—formulation, methodology, and editing: Z.W.; Writing—simulator, experiment, and conclusion: Z.B.; Writing—introduction and related work: X.L.; Writing—critically review: N.Y.; Data curation: X.Y. Management and supervision: H.Z. All authors have read and agreed to the published version of the manuscript.

**Funding:** This research was funded by Beijing Municipal Natural Science Foundation under Grant Agreement Grant No. 4224092 and Scientific Research and Technology Development Project, CNPC (2021DJ7704).

**Data Availability Statement:** The data that support the findings of this study are available from China National Petroleum Corporation (CNPC). Still, restrictions apply to the availability of these data, which were used under license for the current study and are not publicly available. Data are, however, available from the authors upon reasonable request and with the permission of CNPC.

**Acknowledgments:** The authors are grateful for the support of oversight and leadership from Hualin Liu and the project administration and funding acquisition from Lei Yang.

**Conflicts of Interest:** The authors declare they have no financial interest or personal relationships relevant to the content of this article's content.

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
