# Peer review of "Hierarchical Reinforcement Learning for Crude Oil Supply Chain Scheduling"

_algorithms, doi:10.3390/a16070354_

Round 1

Reviewer 1 Report

The paper proposes a method for short-term scheduling for crude oil supply chain based on reinforcement learning. This is a complex problem that could potentially benefit from this sort of approach. This paper is relevant to the journal's readership, as it addresses a pressing issue in the field and offers valuable insights that can significantly impact current research and industry practices. Its contributions are potentially novel and interesting, offering a fresh perspective with the potential to advance knowledge and understanding in this area. The paper is well-written and easy to follow. However, I have two major concerns that would need to be addressed in a revised version:

First, the paper’s contributions need to be stated more precisely and to be better aligned with the literature review. For example, it is unclear how the model they propose for the crude oil supply chain is novel. Modeling by itself is not a contribution. Thus, if the model is stated as a contribution, there should be some novelty that the paper failed to address. Also, presenting numerical results by itself is not a contribution. How does it contribute to the state of the art?

Second, it is unclear how data is being treated to train, validate, and test the proposed HRL model. Is the model tested out-of-sample? Is data partitioned in training, validation, and testing sets? Out-of-sample testing is vital because it evaluates a model's ability to generalize to new, unseen data. It ensures the model can make accurate predictions or decisions in real-world scenarios. By testing on data that was not used during training, it provides an unbiased estimate of the model's performance and helps avoid overfitting. Overall, it plays a key role in validating the reliability and generalizability of models. Furthermore, validation sets are important when evaluating the performance of AI algorithms because they allow for the selection of optimal hyperparameters. By using a separate validation set, which is distinct from the training and test sets, different combinations of hyperparameters can be tested and compared to find the best-performing model. This helps avoid overfitting to the training data and ensures the model's ability to generalize to new data. The validation set acts as a benchmark for evaluating the models' performance and guiding the hyperparameter tuning process. Once the hyperparameters are selected based on the validation set, the final model can be evaluated on an independent test set to assess its generalization ability.

Reviewer 2 Report

The authors are proposing an operational planning/scheduling model for a
crude oil/processed oil supply chain. the problem is important and interesting, and the solution approach is of interest to the readers of the journal. I have a few issues that I need to point out, to make the paper read better:

1. There are language issues throughout the paper, i.e. grammatical, spelling,
semantical, word choice errors that must be fixed. The authors are encouraged to use a professional editing service to remedy this issue.

2. The introduction should be improved. A background of the problem,
motivation for the research and the contributions of the paper must be clearly expressed.

3. Literature review can be improved as well. Instead of saying "OR
approaches," the authors should give more details about the cited work. More references, especially most recent ones should be cited as well.

4. Model description should be improved. It is hard to understand the
mechanism of the solution method. This journal being Algorithms,
pseudocodes and more details are wanted.

5. I don't see any comparison to the existing methods in terms of the solution quality, i.e. in terms of total costs. comparisons are only done on the basis of constraint violations, which is not sufficient. Total cost is the main driver in this problem.

6. Another important aspect of the paper that is missing is the comparison of
the computational effort against the existing approaches. The reader would like to see what is sacrificed in solution quality (by choosing your heuristic) at the expense of what computational gain.

English language editing should be done by a professional editing service, as
the authors' English is hard to understand. There are also numerous spelling,
grammar, and other errors that do not necessarily impede comprehension, but
still make the paper less enticing to read.

Reviewer 3 Report

The paper deals with crude oil resource scheduling. For solving this problem the authors proposed a two-level hierarchical reinforcement learning algorithm. The subject of the paper is interesting and in line with the aims and scope of the Journal. The paper is well-structured and well-written. It provides an original methodology and useful results. However, certain issues need to be addressed.

1.     The abstract is unbalanced. The abstract should provide information on the background, methodology, main results, and main conclusions. It is also recommendable to highlight the main contributions. The background and the methodology in this abstract are dominant. The authors should point out the other elements as well and make the abstract more balanced.

2.     The authors should add one paragraph at the end of the Introduction in which they will shortly describe the following sections of the paper.

3.     Literature review (related work) should be expanded and should include more recent references. Most of the references are older than five years. In addition, the authors should highlight the main research gaps identified based on the related work. What has not been done previously that they will try to bridge in their paper?

4.     The notation of the methodology is extensive. I suggest the authors check again entire notation and if there are any overlaps of already used symbols.

5.     The connection between methodology and problem-solving is not well established. The authors never again mention any of the equations (by the numeration they established) they listed in the description of the methodology. The methodology and problem-solving should be explained in a way to allow the replication of results.

6.     The paper lacks discussion. The authors did not discuss and interpret the results from the perspective of previous studies, nor did they highlight the limitations of the study (proposed model).

7.     The authors did not provide any theoretical or managerial implications of their study.

8.     The future research directions are rather weak. The authors should provide at least 3-5 solid future research directions which would be interesting to the majority of the journal readership.

9.     There are certain technical errors:

-      There should be at least a couple of sentences between headings of different levels (e.g. between section 2 and sub-section 2.1, etc.). This is not mandatory, but it is considered a standard in academic writing.

-      All figures present in the paper must be mentioned somewhere in the main text. For example, figure 8 is not mentioned anywhere in the main text.

-      Table captions should be above the table, not below. Check the caption of Table 3.

-      References in the reference list are not formatted according to the Instructions for Authors (e.g. the journal names are not abbreviated).

-      Page numbers are missing for some references. Check all references.

-      All abbreviations should be defined the first time they appear in the paper. For example, the abbreviation “DQN” is not defined. Check the rest of the paper.

The English language is mostly fine. There are minor style errors that could be addressed. 

Round 2

Reviewer 1 Report

Although the paper has improved, I do not think my second comment was properly addressed in the revision. In Section 5, the authors should clarify extensively in the paper itself how the data was treated and used to conduct the experiments. This is important to determine if the results properly support the conclusions.

Reviewer 2 Report

The authors considered the reviewer's comments and improved the paper.

English language issues are still present, but the paper is readable. For instance, the use/non-use of the articles the/a/an is an issue, which is common for non-English speakers. There is still wrong use of words. For instance, the authors use the word "abstract" in place of "summarize," which is using a noun in place of a verb. The authors misunderstood my comment that called for professional editing service. they responded as "we now use professional words." The paper is readable, but the English can definitely be improved.

Reviewer 3 Report

The authors have invested a substantial effort to address all issues identified in the previous review round, significantly improving their paper's quality. Therefore I propose an acceptance of the paper in its present form.

Round 3

Reviewer 1 Report

I have no further comments